# Novel Insight into the Mechanisms of the Bidirectional Relationship between Diabetes and Periodontitis

**DOI:** 10.3390/biomedicines10010178

**Published:** 2022-01-16

**Authors:** Federica Barutta, Stefania Bellini, Marilena Durazzo, Gabriella Gruden

**Affiliations:** Department of Medical Sciences, University of Turin, 10129 Turin, Italy; federica.barutta@unito.it (F.B.); stefania.bellini@unito.it (S.B.); marilena.durazzo@unito.it (M.D.)

**Keywords:** periodontitis, diabetes, P. gingivalis, inflammation

## Abstract

Periodontitis and diabetes are two major global health problems despite their prevalence being significantly underreported and underestimated. Both epidemiological and intervention studies show a bidirectional relationship between periodontitis and diabetes. The hypothesis of a potential causal link between the two diseases is corroborated by recent studies in experimental animals that identified mechanisms whereby periodontitis and diabetes can adversely affect each other. Herein, we will review clinical data on the existence of a two-way relationship between periodontitis and diabetes and discuss possible mechanistic interactions in both directions, focusing in particular on new data highlighting the importance of the host response. Moreover, we will address the hypothesis that trained immunity may represent the unifying mechanism explaining the intertwined association between diabetes and periodontitis. Achieving a better mechanistic insight on clustering of infectious, inflammatory, and metabolic diseases may provide new therapeutic options to reduce the risk of diabetes and diabetes-associated comorbidities.

## 1. Periodontitis

Periodontitis is a chronic inflammatory disease that affects the periodontium, which includes the gingiva, the periodontal ligament, and the alveolar bone. It is one of the most common chronic oral diseases and a major health burden. Periodontitis has a prevalence of 42% among US adults and, in its severe form, afflicts ~10% of the population, accounting for almost 750 million people worldwide [1,2].

Periodontal disease includes both gingivitis and periodontitis. Gingivitis is a reversible inflammatory process that involves only the gums. Redness, swelling, and bleeding, which is provoked by routine brushing, are characteristic clinical features. Gingivitis can progress to periodontitis that is characterized by a chronic and irreversible damage of soft and hard (alveolar bone) periodontal tissues. Multiple factors, such as genetic susceptibility, microbiota composition, and lifestyle, including cigarette smoking, diet, stress, hormonal factors, can favor the development of periodontitis [3]. Clinical findings include gingival bleeding on probing, increased probing depth, enhanced tooth mobility, and bone loss. If not treated, periodontitis progresses to tooth loss that affects both mastication and aesthetics. Non-surgical periodontal treatment (NSPT) is based on scaling, root planing, and accurate oral hygiene. However, advanced cases need surgical periodontal treatment [3].

Periodontitis is associated epidemiologically with several non-communicable chronic diseases, including obesity, metabolic syndrome (MetS), diabetes mellitus (DM), and cardiovascular diseases (CVD) [4,5,6,7]. Recently, the mechanisms whereby periodontitis may be causally linked to other comorbidities and vice versa have been intensively investigated. Here, we will summarize clinical and epidemiological data on the association between periodontitis and DM and discuss the potential underlying mechanisms of this bidirectional relationship.

## 2. Diabetes Mellitus

DM affects 537 million individuals and this number is expected to reach 783 million in 2045. DM is the seventh cause of death in the US and was responsible for 6.7 million deaths in 2021 [8]. Hyperglycemia caused by either absolute or relative insulin deficiency is the hallmark of DM [9]. In type 1 DM destruction of β cells in the pancreatic islets due to an autoimmune process results in absolute insulin deficiency. Type 2 DM is caused by both peripheral insulin resistance and relative deficiency of β cell insulin secretion. Impaired glucose tolerance and impaired fasting glucose are conditions intermediate between normality and DM and are collectively named pre-DM [9].

Insulin is secreted by β cells and the gut incretin hormones (GLP-1, GIP) enhance glucose-induced insulin secretion. However, incretins are short lived as they are rapidly catabolized by the enzyme dipeptidyl peptidase-4 (DPP4). Once released, insulin binds to the insulin receptor exposed predominantly by the liver, adipose tissue, and skeletal muscle, leading to activation of the insulin-receptor substrate (IRS), which mediates most of the intracellular effects of insulin. Insulin reduces blood glucose levels by inducing hepatic glycogen synthesis via the Akt/GSK-3β pathway, reducing hepatic gluconeogenesis via Akt/FoxO1 inhibition, and increasing peripheral glucose uptake through enhanced exposure of the glucose transporter GLUT-4 on the plasma membrane of peripheral tissues [10,11,12].

Resistance to insulin action (insulin resistance) is the fundamental abnormality of MetS, pre-DM, and type 2 DM. Obesity, predominantly visceral and ectopic obesity, plays a pivotal role in the development of insulin resistance by inducing a chronic low-grade inflammation. Inflammatory cells recruited in the adipose tissue release inflammatory cytokines/adipokines that impair insulin action both locally and systemically. Consistent with this, type 2 DM incidence correlates with circulating levels of inflammatory markers, such as C-reactive protein (CRP), IL-6, TNF-α, chemokines, and white cell count [13,14,15]. Enhanced insulin secretion by β cells can counterbalance the increased resistance to insulin action; however, if β cells fail to compensate for insulin resistance, type 2 DM develops [9].

Glycated hemoglobin (HbA1c) is the most widely used test to assess glucose control in patients with DM. Lowering HbA1c levels is the main strategy to reduce the risk of long-term macrovascular (CVD) and microvascular (retinopathy, nephropathy, neuropathy) complications of DM [9]. Hyperglycemia has direct deleterious effects on target organs of DM complications by inducing oxidative stress, inflammation, and subcellular organelle dysfunction. Hyperglycemia also induces non-enzymatic glycation of proteins, altering their structure, function, and turnover. Moreover, advanced glycation end products (AGEs) that accumulate in various tissues can fuel both inflammation and oxidative stress by binding to their receptor (RAGE) on target cells [16].

## 3. Bidirectional Relationship between Periodontitis and DM

A bidirectional relationship between periodontitis and DM exists independently of associated risk factors and the two diseases additionally affect each other [17]. The increased risk of periodontitis in DM was first described by Löe in 1993, and periodontitis was found as the sixth complication of DM [18]. Since then several epidemiological studies have confirmed that patients with both type 1 and type 2 DM have a three- to fourfold increased risk of periodontitis and severity of periodontitis is much greater in uncontrolled DM [19,20,21,22,23]. Moreover, the American Academy of Periodontology (AAP) and the European Federation of Periodontology (EFP) added DM among the risk factors for the progression of periodontal disease [3,24] and recommend the use of HbA1c for periodontitis grading [3]. Nonetheless, awareness of the risk of periodontitis in DM is still scarce among diabetologists, dentists, and patients [25,26].

The reverse is also true as in non-diabetic subjects periodontitis is associated with higher blood glucose levels and an increased incidence of both pre-diabetes and type 2 DM [20,27,28,29,30]. Of interest, a recent large nationwide prospective population-based cohort study has shown that improving oral hygiene was negatively associated with the occurrence of new-onset DM2 [28]. In addition, in patients with established DM, periodontitis is associated with poorer glycemic control [29,30] and higher prevalence of DM-related complications. Periodontitis is independently associated with CVD, ischemic stroke, neuropathy, nephropathy, and retinopathy [7,25,31,32,33,34,35,36,37]. Moreover, subjects with DM and periodontitis have a greater CVD and all-cause mortality compared to people with DM alone [38].

A recent systematic review pooling data from 53 observational studies confirmed this bidirectional relationship by showing that type 2 DM enhances the risk of developing periodontitis by 34%, while severe periodontitis increases type 2 DM incidence by 53% [20]. Similar results were obtained in another recent meta-analysis that only included prospective studies [39].

Intervention studies have shown that NSPT can decrease HbA1c levels in people with DM and its efficacy is greater in patients with higher baseline HbA1c levels [25,40,41,42,43]. However, most NSPT studies were of short duration (3–6 months) because NSPT could only be delayed for a short period of time in the untreated control group for ethical reasons. Recently, further evidence of efficacy of periodontitis treatment in dysmetabolic conditions came from randomized control trials (RCT) of intensive periodontal treatment. A RCT in patients with MetS and severe periodontitis showed that intensive periodontal treatment is superior to minimal periodontal treatment in reducing circulating markers of inflammation, blood pressure, and HbA1c levels [44]. A 12-month RCT performed in patients with severe periodontitis and type 2 DM demonstrated that intensive periodontal therapy, including surgery, reduced mean HbA1c significantly more (−0.6%) than NSPT [45].

In the opposite direction, improving glucose control is beneficial in preventing both onset and progression of chronic complications of DM and this is also likely true for periodontitis. Moreover, recent studies in both experimental animals and humans suggest that drugs that are currently used for the treatment of DM2 have additional beneficial effects in periodontitis independently of their glucose-lowering action [46,47,48,49,50,51,52]. Local application of 1% metformin gel into the periodontal pockets improves periodontitis when given on the top of scaling and root planning [53]. Moreover, recent meta-analyses concluded that local treatment of periodontitis with metformin gel provides an additional benefit to mechanical periodontal therapy [54,55]. Larger intervention studies are, however, required to establish the effectiveness of specific anti-diabetic agents as adjuvants in periodontitis treatment.

## 4. Pathogenesis of Periodontitis

Periodontitis is the result of an abnormal interaction between the subgingival microbiome and the response of the host [56,57,58].

In normal conditions, the commensal subgingival microbiota is composed predominantly of gram-positive facultative bacteria. These bacteria are enclosed in an extracellular matrix composed mainly of polysaccharides (dental plaque) [59]. If the dental plaque is not removed mechanically, it tends to accumulate, causing gingivitis that can escalate to periodontitis.

In periodontitis, the commensal microbiota shifts to a dysbiotic pathogenic form [60]. Anaerobic gram-negative bacteria become predominant. They are frequently asaccharolytic anaerobic microorganisms that cannot metabolize carbohydrates and depend on peptides and hemin-containing compounds for energy supply [57]. The triad of Treponema denticola, Tannerella forsythia, and Porphyromonas gingivalis (P. gingivalis), called the red complex bacteria, is considered important in the pathogenesis of periodontitis [61]. However, it is dysbiosis that is crucial in the development of periodontitis rather than specific microorganisms [62]. Dysbiosis is defined as “a change in the relative abundance and/or influence of individual components of the bacterial community compared to their abundance in health” [58,62]. For instance, P. gingivalis is present at low levels; however, it has a disproportionately large effect (keystone pathogen) in the pathogenesis of periodontitis as it can turn the commensal microbiota into a dysbiotic microbiota by manipulating the host immune response [58]. Among many other deleterious effects, P. gingivalis blocks complement activation, suppresses both antimicrobial effects and phagocytosis in neutrophils, and inhibits intracellular microorganisms killing in macrophages [63,64,65,66,67,68]. This ensures the survival of P. gingivalis, but also the outgrowth of other microorganisms that can induce pathology (pathobionts). Accordingly, P. gingivalis causes periodontitis in normal animals, but not in germ-free mice, proving that P. gingivalis requires the presence of other bacteria to induce periodontitis [69]. P. gingivalis reduces the host immune response without reducing inflammation as inflammation is important for the dysbiotic microbiota that feed off inflammatory tissue breakdown products [70]. Changes in the microenvironment are also important in dysbiosis development. Both gingivitis-associated inflammation and pocket formation make the local environment anaerobic and enriched with tissue breakdown products that can contribute to the selection of anaerobic gram-negative proteolytic bacteria [56,71].

Bacterial products, such as proteolytic enzymes, contribute to the destruction of the periodontium. Moreover, bacteria fuels inflammation by activating the Toll-Like Receptors (TLRs) and the complement cascade. However, the dysbiotic microbiota is required, but insufficient to induce periodontitis, because it is the inflammatory response of the host to the dysbiotic microbiota that causes periodontal destruction [57]. Consistent with the notion that bacterial dysbiosis will only cause periodontitis in susceptible hosts, there are subjects who do not develop periodontitis despite massive dental plaque accumulation, whereas others with less plaque accumulation are susceptible to the disease. Susceptibility can be related to the host genotype and the presence of a genetic predisposition is supported by studies on twins as well as by familial aggregation of severe forms of periodontitis [72]. Moreover, both comorbidities and environmental factors (stress, diet, smoking) also play an important role.

In susceptible individuals, the inflammatory response triggered by the dysbiotic microbiota is poorly controlled and causes periodontal destruction rather than protection. Massive accumulation of neutrophils in the periodontal pocket is the hallmark of periodontitis [58]. Studies in both human and experimental animals have shown that both diminished and excessive neutrophil recruitment/activity can lead to the spontaneous development of dysbiosis and periodontitis [73,74,75,76]. This proves the importance of neutrophils and also confirms that host factors exert a selective pressure on the local microbiota. Neutrophils contribute to tissue destruction by releasing both metalloproteinases (MMPs) and reactive oxygen species (ROS) [73,76], but they fail to keep under control the dysbiotic microbiota, which penetrates into the connective tissue and induces the production of proinflammatory cytokines, such as TNF-α, IL-1β, IL-23, by interacting with macrophages and dendritic cells (DCs).

IL-23 production by both DCs and macrophages promotes the survival and expansion of proinflammatory Th17 lymphocytes and shifts the Th17/Treg balance in favor of Th17 cells. IL-17 acting on neutrophils and fibroblasts fuels local inflammation and tissue destruction by inducing production of neutrophil chemoattractants, MMPs, and ROS. In turn, recruited neutrophils produce chemokines (CCL2, CCL20) that further recruit Th17 cells [58,77,78,79,80].

Th17-derived IL-17 plays a pivotal role in bone destruction, which is the key event in severe periodontitis. Indeed, IL-17 induces expression in osteoblasts/osteocytes of receptor activator of nuclear factor kappa-Β ligand (RANKL). Binding of RANKL to its receptor on osteoclast precursors induces their maturation into osteoclasts and this results in enhanced alveolar bone reabsorption. Osteoprotegerin (OPG), a soluble decoy receptor of RANKL, limits osteoclast activation. Therefore, the RANKL/OPG ratio is a key marker of alveolar bone resorption in periodontitis. Finally, inflammation reduces production of factors that stimulate osteoblasts/osteocytes to form new bone and thus inhibits osseous coupling [81,82,83].

## 5. Mechanisms Linking Periodontitis and DM

In this section we will describe the mechanisms underlying the two-way relationship between periodontitis and DM (Figure 1).

### 5.1. Periodontitis—Type 2 DM Direction

Studies in animal models of periodontitis have partially clarified the mechanisms whereby periodontitis may increase insulin resistance and lead to glucose intolerance/DM. Oral gavage with human periodontal pathogens and ligature-induced periodontitis (LIP) are two well-established models of periodontitis in experimental animals. Using these models, it has been shown that oral administration of pathogens, such as P. gingivalis and Actinobacillus actinomycetemcomitans, can induce/aggravate insulin resistance and glucose intolerance, particularly in HFD-fed mice [84,85]. Moreover, in Zucker DM fatty rats fed with HFD, LIP accelerates the development of severe insulin resistance [86]. Finally, infection of the periodontal tissue with P. gingivalis, Fusobacterium nucleatum, and Prevotella intermedia also induced periodontitis and aggravated HFD-induced glucose intolerance as well as aggravated HFD-induced insulin resistance [87].

Periodontitis can have systemic effects favoring the development of insulin resistance and DM predominantly by three mechanisms: (1) Dissemination of periodontal bacteria and bacterial products from the periodontal tissues to the bloodstream; (2) induction/magnification of systemic inflammation via spill over of inflammatory cytokines and host response to the dissemination of bacteria/bacterial products; (3) abnormalities in the gut microbiota and increased gut permeability induced by swallowed periodontal bacteria.

#### 5.1.1. Dissemination of Periodontal Bacteria

Bacteraemia has been proven in patients with periodontitis [88] and can be magnified not only by NSPT, but also by tooth brushing and mastication [89,90]. Accordingly, periodontal bacteria and/or bacterial components were found in various extraoral locations [91,92,93,94]. In the bloodstream, P. gingivalis can evade circulating phagocytes by adhering to erythrocyte [95]. Moreover, P. gingivalis can also enter the circulation using monocytes and DC cells as Trojan horses [96], given its ability to survive within them [88,96,97]. Increased susceptibility to certain infections due to altered innate immune response is considered a characteristic feature of DM [98,99,100] and may also favor systemic dissemination of periodontal bacteria in the context of periodontitis. Hyperglycemia, protein glycation, AGEs, oxidative stress, and insulin resistance/deficiency can negatively affect the immune response. Reduced complement activation [101,102], diminished production of cytokines (IFN-γ, IL-2, IL-22) and anti-microbial peptides [103,104,105,106,107], and both neutrophil and macrophage abnormalities [108,109,110] have been reported (see also Section 5.2.3. below). There is also some evidence of an enhanced risk of bacteremia in the context of DM [111]. Moreover, DM-induced endothelial dysfunction and altered microcirculation may also contribute to periodontal bacteria spreading into the bloodstream.

Virulence factors of periodontal bacteria may affect their systemic invasiveness. P. gingivalis has fimbriae that are important for its motility and entry into both local and distant cells. P. gingivalis with type II fimbriae has enhanced local and systemic invasiveness [112]. A longitudinal study on patients with both periodontitis and DM showed an increase rather than a decrease in HbA1c after NSPT in patients with persistence of P. gingivalis with type II fimbriae in periodontal pockets [113], suggesting a relationship between periodontal bacteria virulence and glucose control in DM.

Bacterial virulence factors may directly favor the development/worsening of insulin resistance. P. gingivalis decreased insulin-induced phosphorylation and insulin-induced translocation of FoxO1 in hepatic HepG2 cells, suggesting that P. gingivalis can enhance hepatic glucose output and circulating blood glucose levels by reducing the inhibitory effect of insulin on hepatic gluconeogenesis [114]. Consistently, treatment of obese *db*/*db* DM mice with P. gingivalis deteriorated fasting hyperglycemia by changing intrahepatic glucose metabolism though increased expression of gluconeogenesis-related enzymes [115]. The proteases gingipains of P. gingivalis are associated with the bacterial membrane and also present in the outer membrane vesicles (OMVs) that are released by P. gingivalis. As OMVs can travel to distant sites, they are of particular interest in the relationship between P. gingivalis and DM. In experimental animals, OMVs can translocate to the liver and increase blood glucose levels by inducing hepatic insulin resistance through Akt/GSK-3β-dependent inhibition of hepatic glycogen synthesis [116,117].

Periodontal pathogens might also contribute to insulin resistance through their metabolic activities. Recently, essential branched-chain amino acids (BCAA) have been proposed as a link between P. gingivalis-induced periodontitis and insulin resistance. P gingivalis can produce BCAA and a recent study has shown that worsening of HFD-induced insulin resistance by P. gingivalis-induced periodontitis was paralleled by an increase in BCAA plasma levels. Furthermore, a mutant form of P. gingivalis unable to produce BCAA caused periodontitis, but did not induce insulin resistance [84]. This demonstrates that the BCAA production by P. gingivalis is required for the development of insulin resistance in mice fed with a HFD. Mechanistically, BCAAs are known to activate the mTOR-S6K1 pathway that induces insulin resistance by phosphorylating IRS-1 [118]. Several studies in humans support the relationship between BCAA and insulin resistance [118,119]. There is an association between increased BCAA plasma levels and obesity, insulin resistance, and DM [120,121]. BCAA levels are a predictor of DM onset [122]. Moreover, feeding experimental animals with a BCAA-enriched diet can induce insulin resistance [120] and the gut bacterium P. copri induces insulin resistance and exacerbates glucose intolerance by enhancing BCAA circulating levels [123].

Periodontal bacteria may also affect insulin production. P. gingivalis/P. gingivalis products can translocate to the pancreas and induce changes in islet architecture [124]. In a recent study, P. gingivalis/gingipains was shown to localize primarily in intra- or peri-nuclear areas of the pancreatic β cells in both experimental animals and human pancreatic samples and this was strongly associated with the number of cells with an intermediate phenotype between α and β cells [125]. Based on studies showing that P. gingivalis can induce epigenetic changes [126], it has been proposed that in the nuclei of β cells, P. gingivalis/gingipains may induce epigenetic changes that alters β cell identity by inducing β cell dedifferentiation and thus altered insulin production. GLP-1 regulates islet hormone secretion, glucose concentrations, appetite, body weight, and a downregulation of incretin secretion is observed in subjects with type 2 DM. A recent study has shown that obese non-DM subjects with periodontitis have reduced circulating levels of GLP-1 and a relative increase in glucagone levels [127]. These abnormalities may contribute to impair glucose tolerance and partially explain the higher risk of type 2 DM observed in patients with periodontitis. Of interest, P. gingivalis expresses an active dipeptidyl peptidase 4 (Pg-DPP4) that has 31% homology with human DPP4 [128]. Consistent with the hypothesis that Pg-DPP4 can affect blood glucose levels by enhancing GLP-1 degradation, intravenous injection of Pg-DPP4 decreased both GLP-1 and insulin plasma levels in experimental animals and this was paralleled by a greater increase in blood glucose levels after oral glucose administration compared to control animals [129].

#### 5.1.2. Inflammation

Periodontitis-induced low-grade systemic inflammation is considered an important factor explaining the association between periodontitis and type 2 DM [130].

Bacteria, their products [Lipopolysaccharide (LPS), proteases], and cytokines present in the periodontium can enter into the bloodstream and trigger a systemic inflammatory response in the host. Patients with severe periodontitis have increased levels of pro-inflammatory markers (CRP, fibrinogen, IL-6, IL-1, TNF-α) and neutrophils in the blood [45,88,131] and effective periodontal treatment diminishes these inflammatory markers [44,45,88,132,133]. Similarly, in experimental periodontitis, there is an increase in the circulating levels of CRP, IL-1β, IL-6, and of neutrophils [134,135].

The systemic increase of pro-inflammatory cytokines initiated by periodontal bacteria may promote insulin resistance [4,58]. Consistent with this, amelioration of plasma glucose concentrations and HbA1c levels in response to effective periodontal treatment was paralleled by a reduction in systemic inflammatory markers [44,45]. Moreover, in mice fed with a HFD, oral gavage with P. gingivalis, Fusobacterium nucleatum, and Prevotella intermedia not only exacerbated glucose intolerance and insulin resistance, but also induced inflammation [87].

Among periodontal pathogens, P. gingivalis appears of particular importance as P. gingivalis or P. gingivalis components are sufficient to induce insulin resistance in HFD-fed mice [87,136,137]. Repeated oral administration of P. gingivalis induced endotoxemia [138] and both glucose intolerance and insulin resistance were observed in HFD-fed mice that underwent continuous infusion of P. gingivalis LPS [87]. This suggests that LPS from P. gingivalis is a possible mechanism explaining the link between periodontitis and inflammation-induced insulin resistance. Even very low circulating levels of LPS are sufficient to trigger an inflammatory response via the interaction with TLR2 and TLR4, leading to low-grade systemic inflammation and resulting in insulin resistance. Consistently, endotoxemia induced by oral gavage with P. gingivalis was associated with inflammation in important sites of insulin resistance, such as the liver and the adipose tissue [136]. In the adipose tissue, there was macrophage infiltration with a typical crown-like structure, overexpression of proinflammatory genes (TNF-α, CCL2, IL-6, IL-1) and downregulation of genes that enhance insulin sensitivity [138]. In the liver, there was accumulation of triglycerides, overexpression of genes promoting inflammation (TNF-α, IL-6), lipid droplet formation (Fitm2, Plin2), fatty acid synthesis (Acaca) and gluconeogenesis (G6pc), while IRS-1 was downregulated [138]. Moreover, in the brown adipose tissue infusion of P. gingivalis LPS induced upregulation of inflammation-related genes and downregulation of gene controlling lipolysis (Lipe, Pnpla2), differentiation (Ppar-γ, Adipoq), and thermogenesis (Ucp1, Cidea), indicating that endotoxemia by P. gingivalis can also affect metabolism by altering BAT function [139].

Periodontitis-induced inflammation may also have deleterious effects on β cell function. In DM mice induction of experimental periodontitis worsened not only glucose control, but also glucose-stimulated insulin secretion, indicating β cell dysfunction. In vitro and in vivo data suggests that this occurred via periodontitis-induced IL-12 inhibition of Klotho [140]. Consistently, Klotho is expressed by β cells and Klotho knockout mice show pancreatic islet atrophy, decreased insulin content in the pancreatic islets, and lower serum insulin levels [141]. Besides periodontitis-induced inflammation, the adaptive immune response also plays a role in increasing the risk of DM in periodontitis. In HFD-fed mice colonization of periodontal tissue with P. gingivalis or continuous P. gingivalis-LPS infusion exacerbated insulin resistance and glucose intolerance by inducing an adaptive immune response against P. gingivalis-LPS [87].

#### 5.1.3. Oral-Gut Axis

A recent study has proven that transmission and subsequent colonization of the gut by oral bacteria is a common event in healthy subjects [142]. In periodontitis, swallowed periodontal bacteria can induce gut dysbiosis [58,143] and this is another possible mechanism linking periodontitis and DM [138].

P. gingivalis peptides were found in fecal samples following oral P. gingivalis treatment in db/db DM mice [115]. Moreover, recent metagenomic and proteomic studies showed that oral administration of P. gingivalis changed gut microbiota composition with an increase in Bacteroidales and a decrease in Firmicutes. P. gingivalis-induced gut dysbiosis was paralleled by downregulation of tight junction proteins (TJP-1, occluding) [115,144,145], likely leading to enhanced gut permeability and favoring endotoxemia. Of interest, oral administration of P. gingivalis induced changes in the gut microbiota prior to the development of systemic inflammation, suggesting that periodontitis-induced gut dysbiosis may cause endotoxemia, leading then to systemic inflammation and insulin resistance [143].

The gut microbiota also has several important metabolic activities, including production of vitamins, short-chain fatty acid (SCFA), amino acids, and fermentation of non-digestible substrates. Therefore, periodontitis-induced gut dysbiosis besides favoring endotoxemia may also alter the products of gut bacterial metabolism. Consistently, metabolomic studies have shown differences in the intestinal metabolites of DM mice with and without periodontitis [115]. These intestinal bioactive metabolites are absorbed into the systemic circulation and can thus affect the serum metabolome. Indeed, there were changes in the metabolic serum profiling of mice subjected to oral administration of P. gingivalis, including a rise in both phenylalanine and tyrosine levels [146]. Moreover, in db/db mice oral P. gingivalis administration induced gut dysbiosis, changes in the intestinal metabolites, increased liver gluconeogenesis, and fasting hyperglycemia without changes in the expression of proinflammatory cytokines and insulin resistance, suggesting that besides inducing endotoxemia-related inflammation leading to insulin resistance, periodontitis-induced dysbiosis can also lead to entero-hepatic metabolic derangement by changing the composition of intestinal metabolites [115].

### 5.2. DM-Periodontitis Direction

DM can promote susceptibility to severe periodontitis by contributing to: (1) Periodontal dysbiosis; (2) enhanced inflammatory/immune host response to the bacterial challenge; (3) periodontal tissue destruction. As in other DM complications, hyperglycemia, AGEs, inflammation, and oxidative stress play a key role in mediating these effects.

#### 5.2.1. Changes in the Microbiota

According to the AAP/EFP there is no compelling evidence that DM has a significant impact on the oral microbiota [147]. Available data in humans on the effect of DM on the oral/subgingival microbiota are often inconsistent and contradictory [147,148], likely because of a large number of confounders. Moreover, most available clinical studies have a small sample size and a cross-sectional design. However, a recent longitudinal metagenomic analysis of the subgingival microbiome has shown that among patients who developed periodontitis during follow up, the shift from normality towards dysbiosis was greater in non-DM than in DM subjects, suggesting that DM patients are less tolerant to the presence of periodontal pathogens and that periodontitis can develop in response to a less severe dysbiotic change of the subgingival microbiome [149].

On the other hand, recent in vitro and in vivo studies suggest that DM can increase the pathogenicity of the periodontal microbiota. DM-induced protein glycation promotes the virulence of P. gingivalis by increasing the ability of P. gingivalis to acquire heme from hemoglobin [150]. Animals with type 2 DM showed significant changes in the microbiota composition with a striking reduction in oral microbial diversity compared to non-DM animals [151]. Moreover, transfer of oral bacteria from DM to non-DM germ-free mice induced a greater neutrophil accumulation, IL-6 and RANKL expression, and bone reabsorption compared to transfer from non-DM mice [151], proving that DM can enhance the microbiota pathogenicity. Of interest, blockade of IL-17 by injection of antibodies against IL-17 in the gingiva of DM mice normalized the microbiota composition. Moreover, anti-IL-17 treatment reduced the pathogenicity of bacteria from DM mice. Indeed, transfer of oral bacteria from anti-IL-17 treated DM mice to non-DM germ-free mice had a reduced capacity to induce inflammation, RANKL expression, and bone resorption compared to transfer from untreated DM mice [151]. Taken together, these data indicate that DM causes an increase in IL-17 and inflammation, which alters the periodontal microbiome, favoring an exaggerated pro-inflammatory response, leading to tissue destruction.

#### 5.2.2. Inflammatory Host Response

DM can increase the susceptibility to periodontitis by enhancing the inflammatory response to oral bacteria. Within the periodontium, IL-1, IL-6, and TNF-α levels are enhanced in patients with both DM and periodontitis compared to patients with periodontitis alone and there is a direct relationship between cytokine levels and glucose control [147,152]. Worsening of periodontitis in DM was also observed in experimental animals. DM aggravated periodontal destruction in A. actinomycetemcomitans-induced periodontitis, as shown by a significant increase in TNF expression, leukocyte infiltration, and bone loss [153]. DM rats, which underwent LIP, showed prolonged inflammation [154,155,156] and injection of bacteria in the connective tissue caused greater inflammation in DM than in control animals [157], providing conclusive evidence that DM alters the host inflammatory response to oral bacteria [156,157]. Consistently, in vitro high glucose, AGEs, and P. gingivalis-LPS have synergic effects in modulating TLR expression, NF-kB activation, and proinflammatory cytokine production [158,159,160,161]. A reduction in both anti-inflammatory cytokines (IL-4, IL-10, TGF-β1) [156,162,163] and resolvins [164] may also contribute to enhanced periodontal inflammation. Resolution of inflammation is regulated by proresolving mediators of inflammation, including resolvins [165]. Boosting resolution of inflammation through resolvins diminished DM-induced impairment of P. gingivalis phagocytosis. Moreover, diabetic ERV1 transgenic mice with enhanced activity of resolvins were resistant to periodontitis [164,166]. The DM-induced proinflammatory environment increases vascular permeability, recruitment of inflammatory cells, apoptosis, MMP release, and RANKL expression, resulting in enhanced periodontal destruction [167,168].

Hyperglycemia and AGEs can induce these alterations both directly and indirectly via enhanced mitochondrial oxidative stress [169]. Both glycated albumin and glycated hemoglobin are present in the gingival tissue and the crevicular fluid from periodontal lesions of DM patients. Moreover, increased both AGE deposition and RAGE expression on vessels and monocytes were observed in the gingiva of DM animals with progressive periodontitis [170,171,172,173]. The AGE/RAGE system has been shown to induce an inflammatory response via the NF-κB pathways in vitro in gingival fibroblasts [174] and to increase adhesion of co-cultured monocytic cells [175]. Importantly, blockade of RAGE in DM mice orally treated with P. gingivalis reduced TNF-α and IL-6 production and periodontal bone loss [176]. Furthermore, treatment of LIP-DM rats with aminoguanidine, which prevents AGE accumulation, reduced inflammatory cell infiltration, increased osteoblast number, and reduced osteoclast number [177], confirming the importance of AGE-induced inflammation in coupling DM to periodontal destruction.

Finally, DM can also induce epigenetic changes in the periodontal tissue that may enhance the inflammatory response. Studies in experimental DM have shown changes in DNA methylation levels in the periodontium of more than one thousand genes (599 upregulated and 564 downregulated) in DM animals without periodontitis. Among them, TNF-α and IL-6 genes were hypomethylated and thus more likely to be expressed [178].

#### 5.2.3. Immune Host Response

DM affects the innate and the adaptive immune response, both of which play an important role in the pathogenesis of periodontitis. Hyperglycemia-induced nonenzymatic glycation of complement components reduces complement activation via the lectin pathway and interferes with CD59 that inhibits complement-dependent cytolysis [101,102]. Moreover, DM-induced epigenetic changes have been shown to inhibit both the classical and lectin complement pathways, while the alternative pathway remains active [178]. These abnormalities in the complement cascades may increase the periodontal bacteria pathogenicity without affecting P. gingivalis-induced dysbiosis that occurs via the alternative complement pathway [179].

Studies in both human and experimental DM reported enhanced numbers, but reduced function (chemotaxis, phagocytosis) of neutrophils in the DM periodontal tissue [83]. Poorly functional neutrophils may enhance tissue damage without providing an effective defense against pathogens [180,181,182]. In addition, neutrophils from people with DM overexpress a key enzyme in the formation of neutrophil extracellular traps (NETs) [183]. NETosis has been involved in the pathogenesis of periodontitis [184] and DM-induced enhanced NETosis may promote inflammation and negatively affect the immune defense in the periodontal space.

Macrophages are increased in the DM periodontal tissue and polarized towards the M1 proinflammatory phenotype, while the number of anti-inflammatory M2 macrophages is reduced [83]. This imbalance leading to increased IL-1 and TNF-α release may contribute to exacerbate periodontitis in DM. In keeping with this, injection of M2 macrophages ameliorated P. gingivalis-induced periodontitis [185]. DM-induced changes in systemic metabolism can affect macrophage intracellular cell metabolism that is crucial for macrophage polarization and function. Polarization towards the M1 phenotype requires a shift from oxidative phosphorylation to glycolysis, whereas M2 macrophages primarily rely on oxidative phosphorylation [186]. Changes in macrophage intracellular metabolism during polarization are usually induced by inflammatory cytokines; however, elevated extracellular glucose levels also enhance glycolysis in macrophages and this may itself promote a shift from an anti-inflammatory to a proinflammatory/activated status [187]. In this respect, insulin resistance leading to hyperglycemia may represent a mechanism for assisting the immune system in fighting infections. Besides altering macrophage polarization, DM also enhances myelopoiesis. Indeed, hyperglycemia, particularly intermittent hyperglycemia, induces both proliferation and expansion of bone marrow myeloid progenitors, resulting in an increased release of monocytes in the circulation. Mechanistically, enhanced GLUT-1-mediated glucose uptake and glycolysis in neutrophils induce the secretion of S100a8/a9 that binds to RAGE on bone marrow precursors and induces myelopoiesis [188].

In DM the proinflammatory environment created by macrophages and inflammatory cytokines favors differentiation of naive CD4^+^ T cells in proinflammatory Th17 rather than in regulatory Treg [80]. In addition, the excess of glucose and lipid can also contribute to increasing the Th17/Treg ratio through modulation of nutrient sensing intracellular pathways (mTOR, AMPK, HIF-1) that are crucial in driving differentiation of naive CD4^+^ T cells towards either Th17 or Treg [80]. In turn, Th17/Treg imbalance can further fuel inflammation. As IL-17 plays a key role in periodontitis by both changing periodontal microbiota pathogenicity [151] and driving periodontal bone loss [189], the DM-induced increase in Th17/Treg ratio is likely to contribute to worsening of periodontitis in DM.

#### 5.2.4. Periodontal Tissue Destruction

Potential pathogenic mechanisms for enhanced periodontal tissue destruction in DM include: Diminished generation of collagen, exaggerated collagenolytic activity, enhanced RANKL-mediated osteoclastogenesis, and reduced new bone formation.

Hyperglycemia triggers a variety of collagen alterations. In vivo studies have demonstrated increased collagenase activity and decreased fibroblast collagen synthesis in gingival tissues in both experimental and human DM [190,191,192]. AGEs suppress collagen production by gingival fibroblasts [193] and inhibition of the RAGE/NF-kB pathway rescues high glucose-induced collagen and high glucose-induced fibronectin downregulation in periodontal ligament fibroblasts [194]. In addition, AGEs can modify collagen structure, making the periodontal tissues more susceptible to periodontal breakdown. Neutrophils are the main cellular source of the increased collagenase activity in the gingival crevicular fluid of patients with DM. The collagenase MMP-8 that is the predominant host-cell-derived collagenase that leads to periodontal tissue destruction [195] is increased in gingival samples from patients with DM [196].

In the gingival crevicular fluid of subjects with periodontitis the RANKL/OPG ratio is higher in patients with poorly controlled DM compared to patients with controlled DM [197]. Studies in experimental periodontitis have clarified that DM enhances RANKL expression in osteocytes. Importantly, in this model both bone loss and increased osteoclast numbers/activity were augmented by DM, but completely abolished by osteocyte-specific RANKL deletion in both DM and non-DM animals, demonstrating that inflammation is important, but periodontal bone loss does not occur in the absence of RANKL overproduction by osteocytes [198]. Osteoblasts may also be involved in DM-induced excessive RANKL overexpression [199]. Interestingly, the AGE-RAGE axis has been shown to contribute to osteoclastogenesis by increasing RANKL and downregulating OPG expression [200]. Moreover, enhanced both RANKL production and osteoclast formation in DM were diminished by a TNF-α-antagonist [201].

DM causes a reduction in the number of bone-forming cells by increasing their apoptosis. Studies performed in the LIP model showed that DM induces apoptosis of bone-lining cells, osteoblasts, and periodontal ligament fibroblasts [155,202]. The reduction in fibroblast density following P. gingivalis injury was mediated by TNF-α-induced apoptosis [203]. Moreover, AGEs also induce apoptosis of periodontal ligament fibroblasts via a mitochondrial-dependent mechanism [204].

Periodontal ligament stem cells (PDLSCs) are important in periodontal tissue regeneration. DM can reduce the osteogenic potential of PDLSCs. Indeed, in vitro studies have shown that exposure of PDLSCs to high glucose reduces their proliferation and differentiation towards osteoclasts [205,206].

### 5.3. Bidirectional Relationship between DM and Periodontitis: A Role for Trained Immunity?

Immunological memory is considered a specific feature of adaptive immunity. However, recent studies revealed that microbial/inflammatory factors can elicit a form of memory also in innate immune cells, enabling them to respond more effectively to a second challenge. This innate memory lasting several months has been named “trained immunity” [207,208].

Trained immunity was initially demonstrated in circulating myeloid cells. However, mature myeloid cells are short-lived [209] and cannot provide long-lasting memory. This inconsistency has been recently explained by studies showing that trained immunity also occurs in bone marrow progenitors and is associated with enhanced myelopoiesis [210,211]. Mechanisms of trained immunity have been partially clarified. During the primary challenge, bacterial products and/or inflammatory cytokines trigger changes in cell metabolism of both mature myeloid cells and their progenitors, such as enhanced glycolysis, altered tricarboxylic acid cycle, and reduced mitochondrial oxidative phosphorylation [207]. This leads to accumulation of metabolites, such as fumarate, succinate, mevalonate, acetyl-CoA, that can modulate the activity of chromatin-modifying enzymes, thereby leading to epigenetic changes. The epigenetic rewiring increases chromatin accessibility to genes related to the innate immune response and allows the cell to respond more quickly and robustly to a second unrelated challenge [207,212,213,214,215,216].

Because periodontitis causes systemic inflammation, it is likely to trigger trained immunity in both peripheral cells and precursors. This hypothesis is supported by a recent study on patients with periodontitis who underwent 18F-FDG-PET–CT. The authors found a correlation between metabolic activity within the periodontium, which is a surrogate marker of periodontitis, and hematopoietic tissue activity (a marker of the activity of progenitor cells) [217]. Moreover, circulating neutrophils from patients with periodontitis displayed overexpression of IFN-induced genes and enhanced ROS production when primed with type I IFN [218]. Likewise, exposure of neutrophils and monocytes to either LPS or whole bacteria elicited a much greater production of inflammatory cytokines in cells from patients with periodontitis than in cells from healthy individuals [219] and the effect persisted even after effective periodontal treatment. This suggests that myeloid cells from patients with periodontitis are in a trained state, which enables them to have an enhanced response to a second inflammatory challenge.

Recent work in the field of trained immunity has led to the recognition that not only bacterial components/products and inflammatory cytokines can trigger trained immunity, but also “sterile inflammation”. Oxidized low-density lipoprotein, western diet, and hyperglycemia have been proven to induce trained immunity [220,221,222,223,224,225]. Macrophages obtained from DM patients and kept in culture for six days have an enhanced response to LPS and INF-γ or TNF-α, indicating trained immunity [222]. In macrophages, high extracellular glucose promotes proinflammatory gene expression and macrophage polarization toward a proinflammatory M1 phenotype by enhancing glycolysis. Bone marrow-derived macrophages from DM mice retained these characteristics, even when cultured in normal glucose concentrations, indicating trained immunity induced by hyperglycemia. Moreover, DM-primed macrophages showed an enhanced response to proinflammatory IL-1β. Mechanistically, the priming effect of DM was mediated by epigenetic mechanisms and involved the transcription factor Runx1. Collectively, these new data suggest a direct causal link between hyperglycemia-induced enhanced glycolysis in macrophages and the development of trained immunity [223].

Prolonged hyperactivation of the innate immune system induced by trained immunity may provide explanation for the association between periodontitis and DM. Regardless of whether precursors and circulating innate immune cells are first affected by either periodontitis-induced or DM-induced inflammation, trained immunity can have a deleterious effect on both conditions and may provide a rationale for their bidirectional relationship [58] (Figure 2).

## 6. Conclusions and Perspective

Observational studies provided convincing evidence of a bidirectional relationship between periodontitis and DM. Moreover, intervention trials suggest that periodontal treatment ameliorates both circulating inflammatory markers and HbA1c in patients with DM; however, evidence that successful periodontal treatment can reduce the risk/incidence of type 2 DM is still lacking. Further, large clinical trials should be undertaken to assess the effect of periodontitis treatment on DM and vice versa.

Recent studies in experimental animals suggest that drugs that are currently used for the treatment of DM2, such as metformin, sulfonylureas, and GLP-1 receptor agonists, may also have direct beneficial effects in periodontitis by decreasing inflammation [48,51,52], bacterial dysbiosis [49,50], and bone loss and by enhancing bone formation [47,226]. This suggests that, beyond lowering blood glucose levels, anti-diabetic drug may also have pleiotropic effects of potential relevance in the context of periodontitis.

It is well known that DM affects the microcirculation in the retina, the kidney, and the nerves. However, other vascular beds, including the gingival vessels, may also be involved. Consistently, thickening of the capillary basement membrane, microaneurysms, micro-haemorrhages, reduced gingival blood flow, and enhanced capillary density, suggesting increased angiogenic activity, have been reported [227,228,229,230,231,232,233]. Moreover, a recent meta-analysis showed that the expression of VEGF, a potent pro-angiogenic and vaso-permeabilizing factor, is higher in gingival tissue from DM with periodontitis compared with non-DM subjects with periodontitis [234]. While this area of research has been poorly explored so far, it is of relevance as vascular abnormalities may contribute to increase the risk of periodontitis in DM by amplifying inflammatory processes and altering the periodontal microenvironment. Moreover, in vivo analysis of periodontal microcirculation by videocapillaroscopy may represent a useful tool for the early detection of periodontitis and for the non-invasive assessment of subclinical microangiopathic damage in DM.

Currently, prevention strategies aiming to break the vicious cycle between periodontitis and DM are the mainstay to address the bidirectional relationship between DM and periodontitis. To this end a close collaboration between dentists and diabetologists is required and it is strongly recommended by current guidelines [148,235,236]. The dental team can identify patients at high risk of type 2 DM as well as individuals with undiagnosed type 2 DM. Screening in the dental care setting is important as it allows both patients and diabetologists to take action to prevent the development DM in patients with pre-DM and chronic diabetic complications in patients with established DM. A recent study evaluated the effectiveness DM screening in the dental care setting and found that 31.27% of subjects were at high risk of DM and 15.83% had HbA1c levels in pre-DM or DM range [237]. DM should be considered a risk and a modifying factor for periodontitis and glycemic control used for disease grading. Finally, dentists should treat periodontal disease with the aim of improving not only periodontitis, but also blood glucose control. On the other hand, diabetologists should assess the presence of symptoms and signs of periodontitis [238], refer newly diagnosed patients and patients with suspected periodontitis to the dentists, instructed patients to perform adequate home oral hygiene and to see a dentist regularly [148]. However, implementation of guideline is still challenging because of poor awareness and practical barriers, including the different systems in which dentists and diabetologists work. Therefore, it is important that both dental and medical healthcare professionals inform patients about the relationship between DM and periodontitis and improve inter-professional collaboration.

Any effort should also be made to modify unhealthy habits, such as smoking, obesity, and sedentary lifestyle, in order to reduce the burden of both DM and periodontitis. The key role of visceral obesity in both periodontitis and DM is well established [4,239], though evidence of efficacy of weight loss in preventing/ameliorating periodontitis is still lacking. Physical activity is important in inducing/maintaining weight loss in patients with type 2 DM. Moreover, physical activity has been recently associated with improved periodontal health in patients with type 2 DM [240]. Smoking is considered the most important risk factor for periodontitis as it negatively affects both the natural history of the disease and the response to treatment [241,242,243,244,245,246,247,248]. This effect is predominantly due to reduced gingival perfusion and enhanced susceptibility to infection [249]. Moreover, in patients with DM, smoking accelerates the course of periodontal disease and increases the risk of attachment loss [250]. Therefore, smoking cessation is strongly recommended for improving the general and oral health in people with DM.

In the last decades, a better understanding of the pathogenesis of periodontitis and of the mechanistic links between periodontitis and comorbidities, including DM, has led to the identification of novel potential therapeutic targets. Strategies that modulate the host response, using drugs targeting complement, pro-resolution pathways, trained immunity, and Th17/Treg imbalance, appear promising [251,252] though most of studies were performed in experimental animals and results need to be confirmed in humans. Studies using cutting-edge technologies and integrated multi-omics approaches may further improve our current understanding of the systemic and tissue-specific changes that occur in periodontitis and/or DM and provide additional prevention/intervention strategies in the next future.

Exploring new approaches to periodontitis prevention/treatment may also be relevant for the treatment of associated systemic diseases. Indeed, a better understanding of the intertwined pathogenesis of distinct chronic inflammatory diseases, including periodontitis and DM, may lead to a unifying framework that may explain clustering of inflammatory, infectious, and metabolic diseases and also provide the base for novel therapeutic holistic interventions.

## Figures and Tables

**Figure 1 biomedicines-10-00178-f001:**
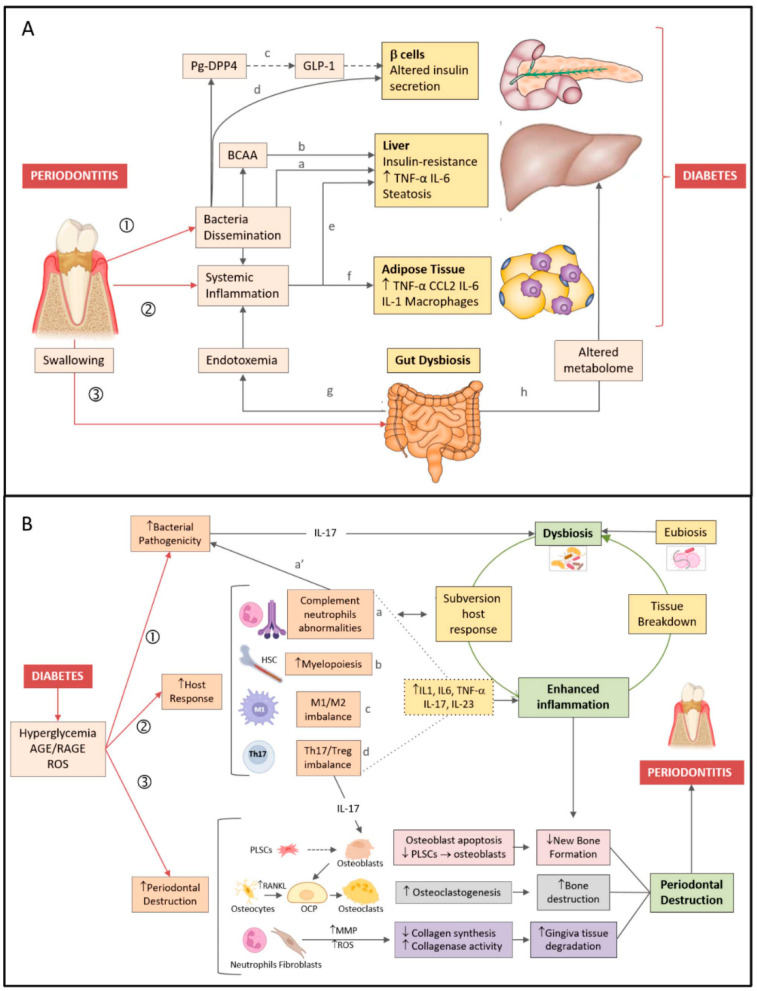
Bidirectional relationship between periodontitis and diabetes. (**A**) Periodontitis diabetes direction. Periodontitis favors development/worsening of type 2 diabetes by three major mechanisms: (1) Dissemination of periodontal bacteria/bacterial products into the bloodstream. Bacteria/bacterial products can induce insulin resistance (a) by inhibiting hepatic glycogen synthesis, increasing hepatic gluconeogenesis, and (b) blocking the insulin receptor substrate via production of branched-chain amino acids (BCAA). (c) Dipeptidyl peptidase-4 (DPP4) produced by P. gingivalis (Pg-DPP4) can reduce glucose-induced insulin production by enhancing glucagon-like peptide 1 (GLP-1) degradation (d) P. gingivalis may alter insulin production by inducing β cell dedifferentiation. (2) Induction/magnification of systemic inflammation, favoring both (e) hepatic and (f) adipose tissue insulin resistance. (3) Gut dysbiosis induced by swallowed periodontal bacteria, favoring both (g) endotoxemia and (h) changes in the blood metabolome. (**B**) Diabetes periodontitis direction. Pathogenesis of periodontitis is depicted on the right hand side of the figure. Dysbiosis, inflammation, and destruction of the periodontium (green boxes) are characteristic features of periodontitis. Dysbiotic bacteria reduce the efficacy of the host immune response, while fuelling inflammation (open green arrow). In turn, inflammation-induced tissue breakdown favors dysbiosis (closed green arrow) closing the vicious cycle. Mechanisms linking diabetes to periodontitis are shown on the left hand side of the figure. Diabetes favors development/worsening of periodontitis by three major mechanisms. (1) Increasing periodontal dysbiosis and bacterial pathogenicity via IL-17. (2) Enhancing the host response to the bacterial challenge. Diabetes (a) alters complement and neutrophil function (which also affects susceptibility to infection a’), (b) increases myelopoiesis, enhances (c) the M1/M2 macrophage ratio, (d) the Th17/Treg lymphocyte ratio, thus raising inflammatory cytokines levels (dotted lines) and fueling inflammation. (3) Increasing periodontal destruction. Diabetes reduces new bone formation by enhancing apoptosis of bone-forming cells and by lowering periodontal ligament stem cells (PLSCs) proliferation and differentiation in osteoblasts (pink boxes). Diabetes enhances osteoclastogenesis by increasing RANKL release by osteocytes/osteoblasts, leading to osteoclast precursor (OCP) differentiation in osteoclasts (grey boxes). Diabetes augments gingiva tissue degradation by increasing release of metalloproteinases (MMP) and reactive oxygen species (ROS) by neutrophils and fibroblasts (violet boxes).

**Figure 2 biomedicines-10-00178-f002:**
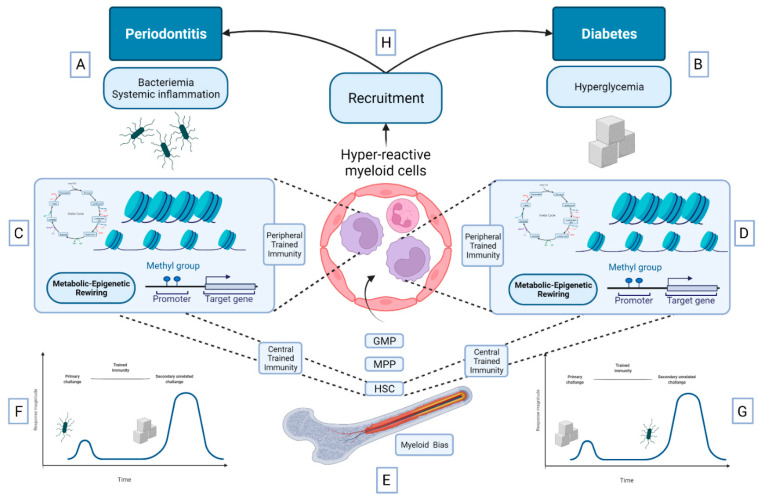
The hypothesis of trained innate immunity as the underlying mechanism of the bidirectional relationship between diabetes and periodontitis. (**A**) Periodontitis-induced release of bacterial products and inflammatory cytokines as well as (**B**) diabetes-induced hyperglycemia may induce metabolic/epigenetic rewiring of both peripheral myeloid cells (peripheral trained immunity) (**C**,**D**) and bone-marrow precursors (central trained immunity) (**E**). This generates hyper-active myeloid cells that can respond more effectively to a second unrelated challenge. (**F**) The graph shows that myeloid cells epigenetically trained by an earlier exposure to periodontitis-related bacterial products may display an enhanced response to hyperglycemia and thus exacerbate diabetes-related inflammation. (**G**) The graph shows that myeloid cells epigenetically trained by an earlier exposure to hyperglycemia may display an enhanced response to bacterial products and thus exacerbate periodontitis-related inflammation. (**H**) Regardless of whether hyperactive myeloid cells are first affected by either periodontitis or diabetes, trained immunity can have a deleterious effect on both conditions and may provide a rationale for their bidirectional relationship. HSC (hematopoietic stem cells), MMP (multipotent progenitors), GMP (granulocyte/macrophage progenitors).

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
