# Peer review of "Novel Insight into the Mechanisms of the Bidirectional Relationship between Diabetes and Periodontitis"

_biomedicines, 2022, doi:10.3390/biomedicines10010178_

Round 1

Reviewer 1 Report

The article titled "Novel Insight on the Mechanisms of the Bidirectional Relation- ship between Diabetes and Periodontitis" is a relevant and interesting update on the relationship between diabetes and periodontitis and vice versa.

I only brief comments with the intention of improving the manuscript.

- If periodontitis is currently the sixth most frequent complication of DM, a more recent bibliographic reference should be sought, otherwise the phrase from a 1993 study should be eliminated.

- It would be interesting to highlight the results of Chang et al (reference 28) that in summary propose that Improving oral hygiene could reduce the risk of occurrence of new-onset diabetes.

- When you first talk about RANKL, I think on line 190, the authors could add the meaning.

- Possibly for the expression Trojan horse, only mention should be made of the reference by Wang et al, although the other references also comment on this expression.

- In the conclusions of the periodontitis-DM topic, it should be emphasized that many of the studies have been carried out in animal models. And if the perspectives are to be highlighted, I would advise new studies under the criteria of prevention rather than therapeutic alternatives. Therefore, I would recommend eliminating the content of reference 206 and 207, although I understand that they are studies to be highlighted.

- If it is possible, make a list of abbreviations

- Correct minor typographical errors in the text: gengiva (line 413), Authors (line 565) ... or spaces of paragraphs not simple.

With the best regards

Author Response

We have appreciated the constructive comments of the Reviewer and addressed them in both our letter and the resubmitted manuscript. Please find our point-by-point responses below. 

Best Regards

Gabriella Gruden

Reviewer 1

The article titled "Novel Insight on the Mechanisms of the Bidirectional Relationship between Diabetes and Periodontitis" is a relevant and interesting update on the relationship between diabetes and periodontitis and vice versa.

I only brief comments with the intention of improving the manuscript.

  • If periodontitis is currently the sixth most frequent complication of DM, a more recent bibliographic reference should be sought, otherwise the phrase from a 1993 study should be eliminated.

We agree with the Reviewer comment and we rephrased the paragraph (lines 87-89).

  • It would be interesting to highlight the results of Chang et al (reference 28) that in summary propose that Improving oral hygiene could reduce the risk of occurrence of new-onset diabetes.

As suggested by the Reviewer, the important finding by Chang et al. showing that improving oral hygiene could reduce the risk of occurrence of new-onset diabetes has been highlighted in the revised version of the manuscript (lines 99-101).

  • When you first talk about RANKL, I think on line 190, the authors could add the meaning.

In the revised version of the manuscript, we explain the RANKL abbreviation the first time it appears in the text (lines 200,201).

  • Possibly for the expression Trojan horse, only mention should be made of the reference by Wang et al, although the other references also comment on this expression.

Only ref 96 (former ref. 87 Wang et al.) was included at the end of the sentence “…using monocytes and DC cells as Trojan horses”.

  • In the conclusions of the periodontitis-DM topic, it should be emphasized that many of the studies have been carried out in animal models. And if the perspectives are to be highlighted, I would advise new studies under the criteria of prevention rather than therapeutic alternatives. Therefore, I would recommend eliminating the content of reference 206 and 207, although I understand that they are studies to be highlighted.

As suggested by the Reviewer, in the new version of the manuscript we acknowledged that new host-based therapeutic strategies for periodontitis treatment, though promising, are based solely on experimental studies. Furthermore, we added a new paragraph to underscore the importance of prevention as well as inter-collaboration between dentists and diabetologists to address the bidirectional relationship between diabetes and periodontitis (lines 662-682; 698-711).

  • If it is possible, make a list of abbreviations

A list of abbreviations has been included in the revised version of the manuscript

  • Correct minor typographical errors in the text: gengiva (line 413), Authors (line 565) ... or spaces of paragraphs not simple.

Spelling and font mistakes have been amended. We apologize for the errors.

Reviewer 2 Report

To achieve a new therapeutic option to reduce the risk of diabetes and diabetes-associated comorbidities, this author discussed the bidirectional association between diabetes and periodontitis. This author also discussed the potential mechanism that might underlying the association between diabetes and periodontitis. As a reviewer, I pointed out same issues that should be discussed in manuscript.

  1. About “Bidirectional relationship between periodontitis and DM” section.

This author discussed the previous studies that reported the beneficial influence of periodontitis therapy on diabetes. However, no study that explain the influence of diabetes therapy on periodontitis. Since this manuscript is intended to showing bidirectional association between periodontitis and diabetes, the information of influence of diabetes therapy on periodontitis is important.

  1. About “Dissemination of periodontal bacteria” section.

Susceptibility to infection is one of the important clinical characteristics for diabetes. And this susceptibility might induce dissemination of periodontal bacteria. Therefore, detail description of susceptibility which relates diabetes should be shown.

  1. In Figure 1-B, this author took neutrophil abnormality as a factor that enhance the inflammation. However, neutrophil abnormality might cause susceptibility to infection that might reduce the inflammation.

  1. In Figure 1-B, neutrophil abnormality might influence on bacterial pathogenicity.

  1. Figures were too complicated to connect the part of figure and part of the description in text. Adding annotation might be helpful for readers.

  1. About “Conclusions and perspective” section.

Even this manuscript intended to discuss about bidirectional association between diabetes and periodontitis, no description about the influence of diabetes therapy on periodontitis was found.

  1. How is the influence of disruption of microcirculation on the association between diabetes and periodontitis?

Nephropathy and retinopathy are well known complications of diabetes. Disruption of microcirculation is pathology of nephropathy and retinopathy. Since disruption of microcirculation also takes important role in developing periodontitis, influence of disruption of microcirculation also should be discussed.

  1. How is the influence of environmental factors both on diabetes and periodontitis?

Habitual status such as smoking and physical activity could influence both on diabetes and periodontitis. And improvement of life style is the main therapy for diabetes. To achieve a new therapeutic option to reduce the risk of diabetes and diabetes-associated comorbidities is main purpose of present review, influence of environmental factors on present association also should be discussed.

Author Response

We have appreciated the constructive comments of the Reviewer and addressed them in both our letter and the resubmitted manuscript. Please find our point-by-point responses below. 

Best Regards

Gabriella Gruden

Reviewer 2

To achieve a new therapeutic option to reduce the risk of diabetes and diabetes-associated comorbidities, this author discussed the bidirectional association between diabetes and periodontitis. This author also discussed the potential mechanism that might underlying the association between diabetes and periodontitis. As a reviewer, I pointed out same issues that should be discussed in manuscript.

  1. About “Bidirectional relationship between periodontitis and DM” section.

This author discussed the previous studies that reported the beneficial influence of periodontitis therapy on diabetes. However, no study that explain the influence of diabetes therapy on periodontitis. Since this manuscript is intended to showing bidirectional association between periodontitis and diabetes, the information of influence of diabetes therapy on periodontitis is important.

As suggested by the Reviewer, we have included a paragraph (lines 124-134) addressing the influence of diabetes therapy on periodontitis. This topic has also been addressed in the “Conclusion and Perspective” Section (see below).

  1. About “Dissemination of periodontal bacteria” section.

Susceptibility to infection is one of the important clinical characteristics for diabetes. And this susceptibility might induce dissemination of periodontal bacteria. Therefore, detail description of susceptibility, which relates diabetes should be shown.

Abnormalities of the immune system in diabetes have been described in the 5.2.3 section (“Immune host response”). However, we agree with the Reviewer that immune system dysfunction, besides affecting the local susceptibility to the development of periodontitis, may also be important in favouring infection and dissemination of periodontal bacteria. Therefore, we added a paragraph to address this relevant point in the “Dissemination of periodontal bacteria” section (lines 265-275).

3.  In Figure 1-B, this author took neutrophil abnormality as a factor that enhance the inflammation. However, neutrophil abnormality might cause susceptibility to infection that might reduce the inflammation.

We agree with the Reviewer that neutrophil abnormalities might increase susceptibility to infection in patients with DM. Figure 1-B has been thus been changed to include an arrow (a’) indicating the possible influence of neutrophil abnormalities on bacterial pathogenicity. The meaning of the arrow was explained in the Legend. On the other hand, both increased susceptibility to infection and enhanced inflammation are characteristic features of both diabetes and diabetes complications, making unlikely that neutrophil abnormalities translate into reduced inflammation in the context of diabetes. Abnormalities of the immune system, enhancing the likelihood of infections, are not always paralleled by a reduction in the inflammatory processes and the effect of Pg. on the host immune system is a good example of the possibility to dissociate the host response to infection from the host inflammatory response.

4. In Figure 1-B, neutrophil abnormality might influence on bacterial pathogenicity.

See above point 3

5. Figures were too complicated to connect the part of figure and part of the description in text. Adding annotation might be helpful for readers.

We agree with the Reviewer observation. Legends to figure have been modified. Moreover, the various parts of figures have been labeled with symbols (alphabetic letters) to clarify the correspondence with the text in the legend.

6. About “Conclusions and perspective” section.

Even this manuscript intended to discuss about bidirectional association between diabetes and periodontitis, no description about the influence of diabetes therapy on periodontitis was found.

As suggested by the Reviewer, we have included a paragraph (lines 642-647) on promising experimental data suggesting an influence of diabetes therapy on periodontitis.

7. How is the influence of disruption of microcirculation on the association between diabetes and periodontitis? Nephropathy and retinopathy are well known complications of diabetes. Disruption of microcirculation is pathology of nephropathy and retinopathy. Since disruption of microcirculation also takes important role in developing periodontitis, influence of disruption of microcirculation also should be discussed.

As suggested by the Reviewer, we have included a paragraph (lines 648-661) on the effect of microvascular dysfunction on periodontitis.

  1. How is the influence of environmental factors both on diabetes and periodontitis? Habitual status such as smoking and physical activity could influence both on diabetes and periodontitis. And improvement of life style is the main therapy for diabetes. To achieve a new therapeutic option to reduce the risk of diabetes and diabetes-associated comorbidities is main purpose of present review, influence of environmental factors on present association also should be discussed.

A paragraph has been added in the “Conclusions and perspective” section to discuss and underscore the importance of environmental factors (obesity, physical activity, smoking) in the bidirectional relationship between DM and periodontitis (lines 683-695).

Round 2

Reviewer 2 Report

I checked this revised version of manuscript. Thanks to this author’s great effort, I think this manuscript has been well improved. Then I think present manuscript folds enough value to be published.